# Ad Hoc Teamwork via Offline Goal-Based Decision Transformers

**Xinzhi Zhang** [1]  **Hohei Chan** [1]  **Deheng Ye** [2]  **Yi Cai** [1]  **Mengchen Zhao** [1]

## Abstract

The ability of agents to collaborate with previously unknown teammates on the fly, known as ad hoc teamwork (AHT), is crucial in many real-world applications. Existing approaches to AHT require online interactions with the environment and some carefully designed teammates. However, these prerequisites can be infeasible in practice. In this work, we extend the AHT problem to the offline setting, where the policy of the ego agent is directly learned from a multi-agent interaction dataset. We propose a hierarchical sequence modeling framework called TAGET that addresses critical challenges in the offline setting, including limited data, partial observability and online adaptation. The core idea of TAGET is to dynamically predict teammate-aware rewards-to-go and sub-goals, so that the ego agent can adapt to the changes of teammates' behaviors in real time. Extensive experimental results show that TAGET significantly outperforms existing solutions to AHT in the offline setting.

## 1. Introduction

Ad hoc teamwork (AHT) refers to enabling autonomous agents to collaborate with unknown teammates without prior coordination (Stone et al., 2010). This capability is crucial in dynamic and uncertain environments where agents must quickly form teams, adapt to new teammates and achieve common goals efficiently (Barrett et al., 2017). For instance, rescue robots should form an ad hoc team in real time to survey affected areas, locate survivors, and deliver supplies. Each robot must adapt to the changes in teammates' policies caused by some unpredictable events. Another typical application of AHT is autonomous driving, where each car must collaborate with random cars nearby to accomplish tasks such as cross-passing and overtaking (Teng et al., 2023).

Previous research on AHT primarily focuses on *online* reinforcement learning (RL) methods (Barrett & Stone, 2015; Durugkar et al., 2020; Mirsky et al., 2020; Ye et al., 2020), which typically require the access to the environment involving various pre-trained teammates. However, these approaches face significant challenges in practice. First, the teammates' joint policy space could be extremely large, and it is impossible to simulate all combinations of teammates' policies during training. Existing works (Gu et al., 2021) usually train the ego agent's policy against a subset of teammates' policies, which greatly reduces the generalization ability of the learned policy. Second, environmental simulators are often unavailable or expensive, especially for real-world environments, without which the online RL approaches cannot be applied. Fortunately, in many cases, offline data collection provides a safer and more cost-effective alternative. For instance, numerous traffic data is recorded by cameras and road sensors every day. This motivates us to study the problem of AHT in an offline setting.

However, addressing AHT problems in the offline setting remains challenging: (1) Offline RL heavily relies on the pre-collected datasets, which can hardly cover diverse teammate behaviors and environmental dynamics, leading to a poor generalization of the learned policy; (2) In partially observable environments, incomplete information makes it more challenging to accurate teammates modeling. This issue is aggravated in the offline setting because we cannot compensate for missing information through active exploration; (3) Existing sequence modeling methods for decision-making (e.g., Decision Transformer (Chen et al., 2021)) primarily rely on the return-guided action generation. However, a simple return signal fails to capture enough information about the frequently changing teammates' intentions and policies in the setting of AHT.

To address the issues mentioned above, we introduce a novel framework called **TAGET**, which stands for **T**eammate-**A**ware **G**oal driven hi**E**rarchical Decision **T**ransformers. Specifically, TAGET adopts a hierarchical framework consisting of a high-level module for teammate-aware sub-goal prediction and a low-level module for action generation. The key innovations of TAGET are as follows. (1) We adopt a *trajectory mirroring* strategy to improve data efficiency, where each agent plays as the ego agent in turn, so that one piece of multi-agent trajectory can be used multiple times

[1]School of Software Engineering, South China University of Technology, Guangzhou, China [2]Tencent, Shenzhen, China. Correspondence to: Mengchen Zhao <zzmc@scut.edu.cn>.

*Proceedings of the 42nd International Conference on Machine Learning*, Vancouver, Canada. PMLR 267, 2025. Copyright 2025 by the author(s).

for training; (2) We employ two separate state encoders for the ego agent and all the agents respectively. By regularizing the output of the two encoders, the ego agent is able to infer the teamwork information based only on its local observations; (3) We extend the conventional return-to-go (RTG) to teammate-aware return-to-go (TA-RTG), which accurately reflects the team return-to-go conditioned on the current global state. Based on the TA-RTG, a teammate-aware sub-goal (TA-Goal) is decoded and used to guide the low-level Decision Transformer to generate corresponding actions. By dynamically predicting the TA-RTG and the TA-Goal at every step, the ego agent is able to adapt to the changes of teammates in real time. The main contributions of our work are summarized as follows.

1. We extend the AHT problem to the offline setting, where an ego agent needs to learn a policy that adapts to various teammates from the offline dataset.

2. We propose a hierarchical sequence modeling framework called TAGET, which successfully addresses the offline AHT problem using a teammate-aware goal-conditioned Decision Transformer.

3. We empirically validate TAGET in three classic environments. The results show that TAGET achieves the best performance against strong baselines, with an average improvement of 37.83% across all environments.

## 2. Related Work

**Ad hoc teamwork.** Existing works on AHT predominantly rely on online interaction with the environment and associated teammates. Early approaches assume fully observable environments with a fixed small number of teammates, and train an agent to collaborate with them through plenty of online interactions (Barrett et al., 2017; Chen et al., 2020; Rahman et al., 2021b; Huang et al., 2021). Recent works consider more realistic settings. For example, ODITS (Gu et al., 2021) and LIAM (Papoudakis et al., 2021) improve the adaptability of the ego agent in partially observable environments. GPL (Rahman et al., 2021a) and OAHT (Wang et al., 2024) learn to address the open ad hoc problem, where the number of teammates can change. TEAMSTER leverages model-based RL methods to approximate the potentially complex environments and teammates (Ribeiro et al., 2023). However, these online approaches perform poorly in the offline setting, because they heavily rely on online explorations to estimate the environments and teammates.

**Offline reinforcement learning.** Offline RL (Levine et al., 2020) aims to learn policies from static datasets without online exploration, effectively reducing data costs and safety risks (Xu et al., 2022a), with notable success in single-agent settings (Qu et al., 2023; Wei et al., 2021). Traditional offline RL algorithms, such as Batch-Constrained Q-learning

(BCQ) (Fujimoto et al., 2019) and Conservative Q-learning (CQL) (Kumar et al., 2020), focus on addressing the out-of-distribution (OOD) problem by constraining policy learning. Decision Transformer (DT) (Chen et al., 2021; Li et al.; Xie et al., 2023) and Trajectory Transformer (TT) (Janner et al., 2021; Xu et al., 2021) reframe offline RL as a sequence modeling problem, achieving performance comparable to state-of-the-art offline RL methods while providing a novel perspective on policy learning (Zheng et al., 2022; Xu et al., 2022b; Reed et al., 2022; Badrinath et al., 2024; Ma et al., 2023). Furthermore, recent works (Yang et al., 2021; Meng et al., 2021; Tseng et al., 2022; Jiang & Lu, 2023) have extended offline RL to multi-agent settings and tried to learn an optimal joint policy from the offline datasets. However, these works do not apply to the ad hoc setting, where the teammates' policies may not follow the optimal joint policy. By contrast, we try to learn a teammate-aware policy that can adapt to the changes to teammates' policies in real time.

**Teammate modeling.** Teammate modeling (Barrett et al., 2012) greatly facilitates multi-agent learning by helping the agents to understand each other. Traditional AHT methods primarily rely on accurate teammate modeling to adapt to unknown teammates, which refines teammate models in real time through online interaction using approaches such as meta-learning (Al-Shedivat et al., 2017; Kim et al., 2021), Bayesian inference (Zintgraf et al., 2021), and value decomposition (Zhang et al., 2023; Gu et al., 2021). While these methods perform well in online settings, they struggle in offline settings due to the inability to constantly improve the teammate models by interacting with the environment (Ma et al., 2024). Although some offline teammate modeling approaches leverage methods like in-context learning (Jing et al., 2023) to compensate for the inability to learn from online interactions, they still struggle with low accuracy. In our work, instead of predicting each teammate's policy, we directly predict the future global states, which are more informative to the ego agent as they effectively summarize the influence of teammates' policies on the environment.

## 3. Problem Statement

Our objective is to train an ego agent to collaborate with unknown teammates in partially observable environments from a fixed dataset. To this end, we formulate the problem as a decentralized partially observable Markov decision process (Dec-POMDP) with an additional joint policy space of teammates, which can be formally described as a tuple $\langle N, \mathcal{S}, \mathcal{O}, \mathcal{A}, \mathcal{T}, P, R, O, \Gamma \rangle$, where $N = \{1, 2, \ldots, n\}$ represents the set of agents involved in the task and $\mathcal{S}$ denotes the global state space. We consider a partially observable setting in which each agent receives an individual observation $o^i \in \mathcal{O}$ via an observation function $O_i : \mathcal{S} \times \mathcal{A} \mapsto \Delta(\mathcal{O}_i)$. The joint action space is defined as $\mathcal{A} = \mathcal{A}^1 \times \mathcal{A}^2 \times \cdots \times \mathcal{A}^N$.

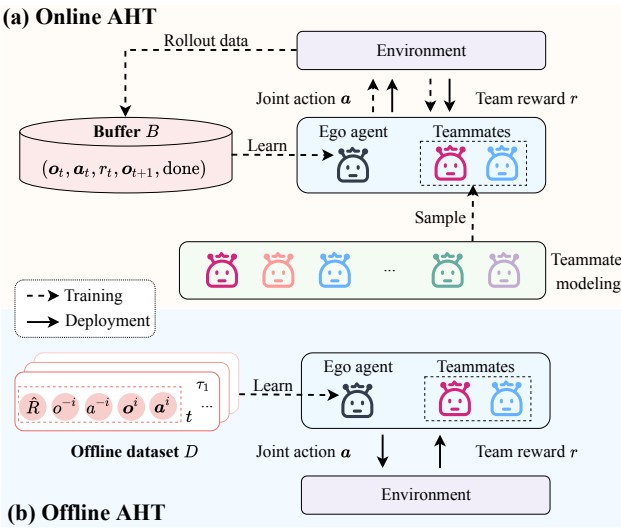

*Figure 1.* Comparison of Online and Offline Ad Hoc Teamwork Frameworks. (a) shows the online AHT framework, where the ego agent learns through real-time interaction, collecting data and updating its policy continuously. (b) illustrates the offline AHT framework, where the agent trains on a pre-collected dataset.

$\tau^i \in \mathcal{T} \equiv (\mathcal{O} \times \mathcal{A})^*$ represents the observation-action trajectory of agent $i$. The transition function $P : \mathcal{S} \times \mathcal{A} \mapsto \Delta(S)$ determines the probability distribution over the next state, given the current state and joint action. In cooperative scenarios, all agents share the common reward function $R : \mathcal{S} \times \mathcal{A} \mapsto \Delta(\mathbb{R})$. We denote by $\Gamma$ the joint policy space of teammates. In practice, $\Gamma$ can be approximated by sampling from a pre-collected diverse policy pool.

Figure 1 presents the comparison between online AHT and offline AHT. Unlike online methods, where agents interact with the environment during training, our approach uses an offline dataset $D$ to train an ego agent policy $\pi_\theta^i(a^i|o^i; D)$. The trained ego agent is then deployed with unknown teammates, whose policies are randomly sampled from the joint policy space $\Gamma$. The goal for the ego agent is to achieve the maximum expected return:

$$\max_\theta \ \mathbb{E}_{\boldsymbol{\pi}^{-i} \sim \Gamma, \mathcal{D}} \left[ \sum_{t=0}^\infty r_t \ \Big| \ a_t^i \sim \pi_\theta^i, \boldsymbol{\pi}^{-i} \right]. \quad (1)$$

## 4. Method

The offline AHT problem faces many practical challenges, including limited data, inaccurate teammate modeling in partially observable environments, and online adaptation to previously unknown teammates. In this section, we introduce a hierarchical goal-based Decision Transformer framework called TAGET, which addresses the above challenges by a unified sequence modeling approach.

### 4.1. Overview

TAGET aims to train an ego agent from an offline dataset so that it can maximize the expected team reward given any teammates' policies. Figure 2 illustrates the framework of TAGET, which comprises three main modules. First, the offline data pre-processing module mitigates the issue of limited data through a trajectory mirroring strategy, where each agent alternates as the ego agent to make the most use of multi-agent trajectory data. Second, the high-level teammate-aware sub-goal prediction module learns a latent variable from local observations to capture the core information of teamwork situations. Specifically, it predicts teammate-aware return-to-go and then combines it with the latent variable to predict teammate-aware sub-goal, which guides decision making and improves the adaptation to dynamic teammate behaviors. Finally, the low-level goal-conditioned action generation module utilizes the Decision Transformer to predict actions conditioned on the high-level TA-Goal, ensuring flexibility and robustness in non-stationary environments.

### 4.2. Offline data collection and pre-processing

As illustrated in Figure 2, to enable policy learning without online interaction, we first collect a comprehensive offline dataset of multi-agent interaction trajectories, denoted as $\mathcal{D} = \{\tau_1, \tau_2, \ldots, \tau_M\}$. Each trajectory $\tau_j$ represents a complete episode of interaction and is structured as follows:

$$\tau_j = \{\hat{R}_1, \boldsymbol{o}_1, \boldsymbol{a}_1, \ldots, \hat{R}_T, \boldsymbol{o}_T, \boldsymbol{a}_T\}. \quad (2)$$

Here, $\mathbf{o}_t$ denotes the joint observations, $\mathbf{a}_t$ denotes the joint actions, and $\hat{R}_t = \sum_{t'=t}^T r_{t'}$ represents the return-to-go at time step $t$. Such offline datasets are relatively easy to collect in real-world scenarios, as they can be obtained by recording interactions during task execution.

Note that, the raw multi-agent interaction trajectories treat all agents equally without differentiating the ego agent and the teammates. If any agent is arbitrarily selected as the ego agent, it would ignore the decision-making perspective of other agents as the ego agent. Therefore, we propose a data pre-processing technique called trajectory mirroring strategy, which reorders the positions of states and observations within a trajectory so that each agent alternates as the ego agent. This strategy effectively utilizes the dataset to extract information from multiple decision-making perspectives and mitigates the challenge of limited data.

Formally, for a trajectory $\tau_j$ with $N$ agents, the trajectory mirroring strategy generates $N$ trajectories by re-ordering the positions of the states and observations. For each mirrored trajectory $\tau_{j,i}$, agent $i$ is treated as the ego agent, while the remaining agents are considered as teammates.

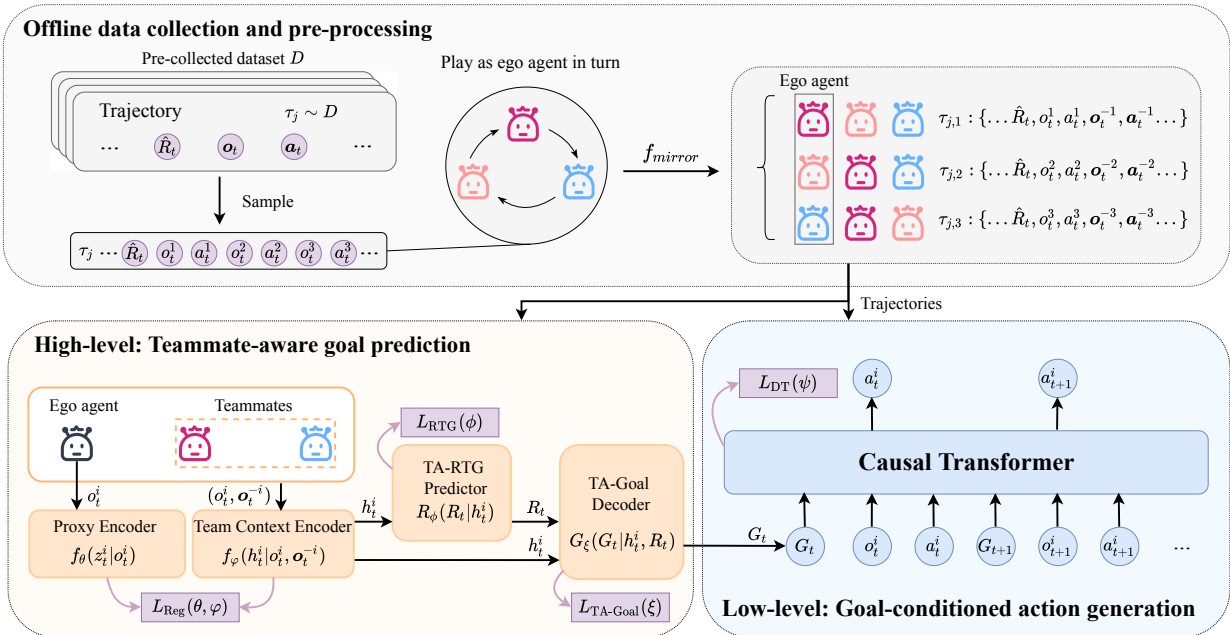

*Figure 2.* Training overview of the proposed TAGET. First, the (1) Offline data collection and pre-processing module collects trajectories and then generates mirrored trajectories by treating each agent as the ego agent. These processed trajectories are then learned by the hierarchical decision framework. At each time step $t$, the (2) High-level: Teammate-aware goal prediction module extracts a latent representation $z_t^i$ from local observation $o_t^i$, approximating the team context $h_t^i$ by regularization. The module predicts TA-RTG $R_t$ based on $h_t^i$ and then utilizes $R_t$ and $h_t^i$ to predict TA-Goal $G_t$. Finally, the (3) Low-level: Goal-conditioned decision-making module employs a Causal Transformer to condition on the predicted goal $G_t$, generating the next action for the ego agent.

The transformation is defined as:

$$\tau_{j,i} = f_{\text{mirror}}(\tau_j, i) = \{\hat{R}_t, o_t^i, \boldsymbol{o}_t^{-i}, a_t^i, \boldsymbol{a}_t^{-i}\}_{t=1}^T, \quad (3)$$

where $o_t^i$ and $a_t^i$ are the observation and action of the ego agent, while $\boldsymbol{o}_t^{-i}$ and $\boldsymbol{a}_t^{-i}$ represent the joint observations and actions of all the other agents. For example, as shown in Figure 2, if a trajectory involves three agents, the trajectory mirroring strategy generates three trajectories: $\tau_{j,1}$, $\tau_{j,2}$, and $\tau_{j,3}$. These trajectories are distinct in downstream modules because the information of the ego agent and the teammates is processed in different ways.

### 4.3. High-level: teammate-aware goal prediction

For decision making, vanilla DT relies on a scalar return-to-go (RTG) signal to guide policies. However, teammates may exhibit highly dynamic and unpredictable behaviors in AHT, so a simple RTG can hardly capture uncertainties in teammates and the environment. To this end, we introduce a new concept called teammate-aware sub-goal (TA-Goal), the predicted global state at a future step that serves as a high-level objective. TA-Goal captures the dynamics of teammate behaviors, providing more robust and adaptive guidance than RTG in complex multi-agent settings. To predict the TA-Goal, we use the team context and

the teammate-aware RTG (TA-RTG) as inputs. The team context encodes the current global state, while the TA-RTG estimates the expected return. Taking these two as inputs, the TA-Goal prediction captures both the current state and future return, serving as the high-level objective to guide the ego agent action generation.

First, we learn to represent team context under partial observability. In multi-agent scenarios, each agent typically has access only to local observations, which hinders understanding of teammates' intentions. We propose two state encoders: team context encoder for all agents and proxy encoder for the ego agent. With the regularization between the outputs of the two encoders, the ego agent is able to infer the team context based only on its local observations.

**Team context encoder:** The team context is a high-level semantic representation derived from the current observations of all agents. To model this, we represent the team context in a stochastic embedding space $\mathcal{H}$, where it is encoded as a latent probabilistic variable $h^i$ drawn from a multivariate Gaussian distribution. We utilize a function $f$ to learn the parameters of the multivariate Gaussian distribution:

$$(\mu_{h^i}, \sigma_{h^i}) = f_\varphi(o_t^i, \boldsymbol{o}_t^{-i}), \quad h^i \sim \mathcal{N}(\mu_{h^i}, \sigma_{h^i}). \quad (4)$$

**Proxy encoder:** Due to the partial observability during execution, directly applying $f$ to capture the team context is infeasible. To overcome this limitation, we introduce a Proxy Encoder $f^*$ that approximates the team context from only local observations $o_t^i$, assuming that $o_t^i$ can partially capture the cooperative context as it inherently reflects teammate interactions and behavioral features. Similarly, $z^i$, as an approximation of $h^i$, is encoded into a stochastic embedding space, where it is represented as a latent probabilistic variable drawn from a multivariate Gaussian distribution. The Proxy Encoder $f^*$ is used to learn the parameters of this Gaussian distribution:

$$(\mu_{z^i}, \sigma_{z^i}) = f_\theta(o_t^i), \quad z^i \sim \mathcal{N}(\mu_{z^i}, \sigma_{z^i}). \quad (5)$$

Here, $z^i$ serves as a latent representation of teammate dynamics, robustly guiding decision-making when global observations are unavailable.

**Latent variable regularization:** To ensure $z_t^i$ to effectively represent the team context under partial observability, we aim for $z_t^i$ to remain consistent with $h_t^i$. Inspired by previous works (Ajay et al., 2020; Pertsch et al., 2021; Xie et al., 2023), we use the following objective to minimize the discrepancy between the two representations and stabilize the high-level representation:

$$L_{\text{Reg}}(\theta, \varphi) = \mathbb{E}_{\tau \sim \mathcal{D}} \left[ \sum_{t=1}^{T} D_{KL} \left( f_\varphi(\cdot \mid o_t^i, \boldsymbol{o}_t^{-i}) \| \mathcal{N}(0, I) \right) \right]$$
$$+ \beta \mathbb{E}_{\tau \sim \mathcal{D}} \left[ \sum_{t=1}^{T} D_{KL} \left( \lfloor f_\varphi(\cdot \mid o_t^i, \boldsymbol{o}_t^{-i}) \rfloor \| f_\theta(z_t^i \mid o_t^i) \right) \right], \quad (6)$$

where $D_{\text{KL}}$ denotes the Kullback-Leibler (KL) divergence, $\lfloor \cdot \rfloor$ indicates the stop-gradient operator, and $\beta$ is a hyperparameter balancing the two regularization terms. The first KL-divergence term $D_{\text{KL}} \left( f_\varphi(\cdot \mid o_t^i, o_t^{-i}) \| \mathcal{N}(0, I) \right)$ is designed as a reverse KL to enforce a compact latent space. Following the beta-VAE framework (Higgins et al., 2017), reverse KL encourages the learned latent distribution to concentrate around the modes of the prior standard normal distribution, effectively preventing overfitting to noisy observations and improving generalization to unseen teammates. In contrast, the second term employs forward KL, which aligns the proxy encoder's output with the team context encoder by minimizing the divergence in expectation. This hybrid regularization ensures the latent space regularity and the consistency between local and global representations under partial observability.

Then, we learn to predict TA-RTG, which is conditioned on the team context. Unlike traditional RTG in single-agent settings, TA-RTG captures the influence of dynamic teammates, making it better suited for AHT scenarios. This enables a more accurate and adaptable estimate of future performance and informs the TA-Goal generation process.

**Teammate-aware return-to-go (TA-RTG):** The TA-RTG $R_t$ represents the predicted return-to-go based on the team context $h_t^i$, which quantifies the contribution of teamwork situation to long-term task impact. It provides critical dynamic optimization feedback for generating TA-Goal. To predict the TA-RTG, we design an RTG predictor $R_\phi$ that takes $h_t^i$ as input: $R_\varphi(R_t | h_t^i)$. The training objective of the predictor is to minimize the discrepancy between the predicted RTG $R_t$ and the ground truth $\hat{R}_t$. Specifically, we use the following MSE loss function:

$$L_{RTG} = \mathbb{E}_{\tau \sim \mathcal{D}} \left[ \sum_{t=1}^{T} \left( R_\phi(h_t^i) - \hat{R}_t \right)^2 \right], \quad (7)$$

Although the TA-RTG provides more accurate information than conventional RTG by considering teammates' behaviors, it is still less informative for downstream action generation, especially in environments with sparse rewards. Therefore, we further predict the TA-Goal, which is represented by the future global state and provides richer contextual information than the TA-RTG. Based on the team context and the TA-RTG, we predict the TA-Goal to guide the decision of the ego agent. The TA-Goal enables the ego agent to coordinate its actions with dynamic teammate behaviors, thereby enhancing robust collaboration in complex multi-agent environments.

**Teammate-aware goal (TA-Goal):** The TA-Goal $G_t$ represents the predicted global state at time $t + k$, which provides a high-level objective to guide low-level decision-making. It is predicted based on the team context $h_t^i$ and the predicted TA-RTG $R_t$. To predict the TA-Goal, we designed the TA-Goal decoder $G_\xi$ that takes $h_t^i$ and $R_t$ as inputs: $G_\xi(G_t | h_t^i, R_t)$. The TA-Goal decoder is trained by minimizing the difference between the predicted $\hat{G}_t$ and the ground-truth $G_t$ in the dataset. $G_t$ is constructed by concatenating the discretized and one-hot encoded observations of all agents at time step $t + k$, representing the global state at that future moment. The loss function is defined as:

$$L_{\text{TA-Goal}} = \mathbb{E}_{\tau \sim \mathcal{D}} \left[ -\frac{1}{N} \sum_{t=1}^{T} \sum_{i=1}^{N} \left( G_{t,i} \log \hat{G}_{t,i} \right. \right.$$
$$\left. \left. + (1 - G_{t,i}) \log(1 - \hat{G}_{t,i}) \right) \right], \quad (8)$$

where $G_{t,i} = s_{t+k} = \text{Concat}\left( \{o_{t+k}^i\}_{i=1}^n \right)$. In total, the overall loss function $L_{\text{High-level}}$ for the high-level network is defined as:

$$L_{\text{High-level}} = \lambda_{\text{Reg}} L_{\text{Reg}} + \lambda_{\text{RTG}} L_{\text{RTG}} + \lambda_{\text{TA-Goal}} L_{\text{TA-Goal}}, \quad (9)$$

where $\lambda_{\text{MI}}, \lambda_{\text{RTG}}, \lambda_{\text{TA-Goal}}$ are weighting parameters used to balance the contributions of different loss terms.

## 4.4. Low-level: Goal-conditioned action generation

We use a goal-based Decision Transformer as the low-level policy network and solve the offline AHT problem through conditioned sequence modeling. The context for the low-level module is composed of two inputs: the TA-Goals predicted by the high-level network and the pre-processed trajectory segments. Specifically, the inputs to the low-level DT are constructed as follows:

$$\tau_{j,i}^* = \{G_{t'}, o_{t'}^i, a_{t'}^i\}_{t'=t-K+1}^t, \tag{10}$$

where $K$ denotes the context length, i.e., the number of past time steps included in the sequence. The causal Transformer processes this sequence and outputs the predicted action $a_t^i$ for the current time step. The training of the low-level module aims to minimize the deviation between the predicted action and the ground-truth action from the dataset. The loss function for training is defined as:

$$L_{\text{DT}}(\psi) = \mathbb{E}_{\substack{\tau \sim \mathcal{D} \\ G_t \sim G_\xi}} \left[ -\log P_\psi \left( a_t^i \mid \tau^{1:t-1}, o_t^i, G_t \right) \right]. \tag{11}$$

We replace RTG with TA-Goal as the prompt to overcome the limitations of DT in AHT problems. Compared to RTG, TA-Goal empowers the model to take actions based on the future state adapted to dynamic teammate behaviors. This allows the model to effectively collaborate with unknown teammates in dynamic MARL environments.

## 5. Experiments

### 5.1. Environments and Data Collection

To evaluate the effectiveness of our proposed method, we conducted experiments in three widely used multi-agent reinforcement learning (MARL) environments: **Predator-prey (PP)**, **Level-Based Foraging (LBF)** and **Overcooked**. These environments allow manipulation of teammates' policies, thus are suitable for evaluating AHT methods. The Predator-Prey environment is a classic MARL benchmark where a team of predators must collaborate to capture prey in a partially observable grid world(Lowe et al., 2017). Successful teamwork requires coordination, as the prey can evade capture without a joint effort. In the LBF environment, agents are tasked with collecting food items scattered randomly in a grid-world. Each agent and food item is assigned a level and a group of agents can collect an item only if the sum of their levels meets or exceeds the level of the item (Papoudakis et al., 2020). The Overcooked environment requires agents to prepare and deliver dishes efficiently in complex kitchen layouts (Carroll et al., 2019). Detailed information about the environments is provided in the Appendix A for further reference.

To train our model in an offline setting, we utilize pre-collected interaction trajectories. To ensure the model's adaptability to diverse teammate strategies, we adopt the Soft-Value Diversity (SVD) method proposed in CSP (Ding et al., 2023) to collect data. This method encourages the inclusion of a wide range of behavioral policies in the dataset. We trained four distinct populations of multi-agent reinforcement learning (MARL) policies for each environment. From these, one population was randomly sampled as the testing teammate set, while the remaining three were used to collect interaction trajectories for the offline dataset. Each training population consists of three unique checkpoints, while each testing population includes eight checkpoints. Each checkpoint represents a distinct joint strategy, capturing a diverse range of behaviors and interaction patterns. By separating training and testing populations, this approach ensures a rigorous evaluation of the model's ability to adapt to unseen teammate policies.

To enhance the coverage of offline datasets, we employed a trajectory mirroring strategy, as we suggested above. For each trajectory in the dataset, this method treats every agent as the ego agent in turn, generating additional trajectories from different perspectives. The resulting dataset enables more robust policy learning in dynamic and partially observable environments, as demonstrated in our experiments.

### 5.2. Baselines

We compare our method against several state-of-the-art baselines commonly used in offline reinforcement learning and ad hoc teamwork tasks: **Decision Transformer (DT)**, **Prompt-DT**, **ODITS-off** and **LIAM-off**. These baselines are representative as they address the AHT problem from different perspectives.

- **Decision Transformer (DT)** (Chen et al., 2021) reformulates reinforcement learning as a sequence modeling task, where the policy is learned by predicting the next action given a sequence of observations, actions and return-to-go. DT has demonstrated strong performance in offline reinforcement learning settings by leveraging the transformer architecture to model long-term dependencies in sequential data.

- **Prompt-DT** (Xu et al., 2022b) is an extension of DT that incorporates additional prompts to guide the decision-making process to achieve a few-shot adaptation. This method has been shown to enhance generalization and adaptability, making it an appropriate baseline for our approach, which also deals with diverse and dynamic teammate behaviors.

- **ODITS-off** is a variant of the ODITS framework (Gu et al., 2021) which uses mutual information regularization to infer teammates' behaviors from local observation. We treat the trajectories in the offline dataset as if they were obtained through online interactions,

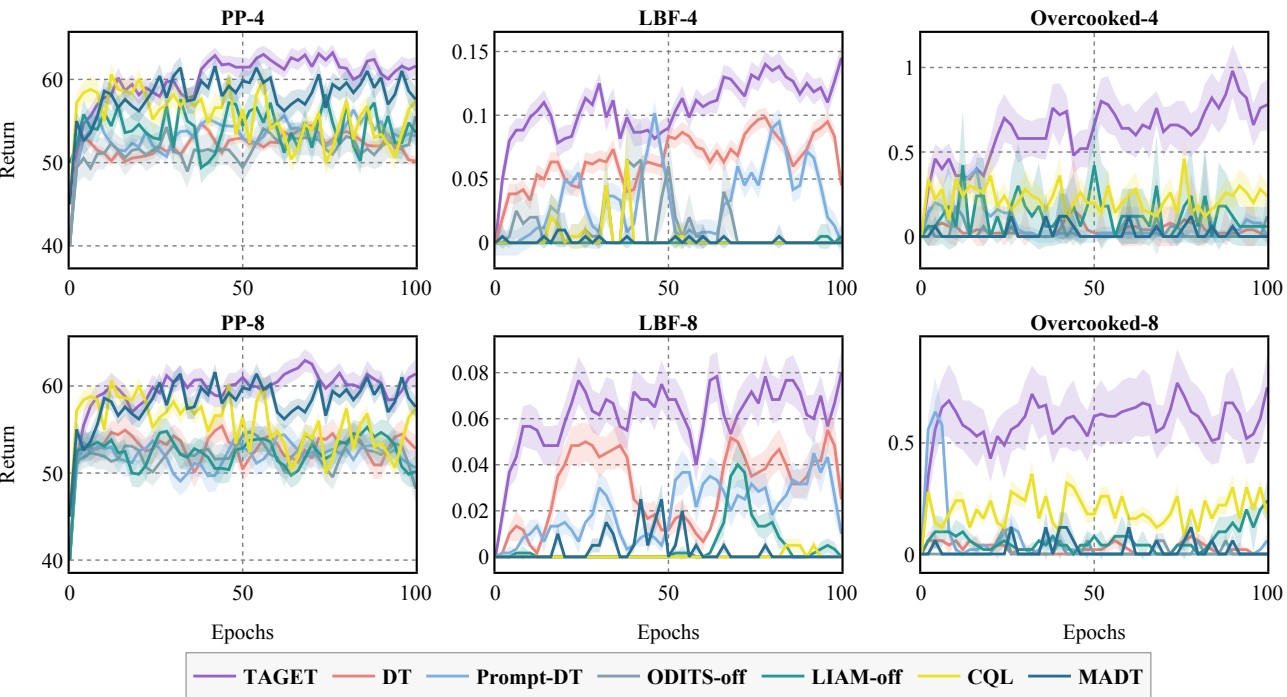

Figure 3. Average evaluation returns curves and 95% confidence interval of the five evaluated methods during training

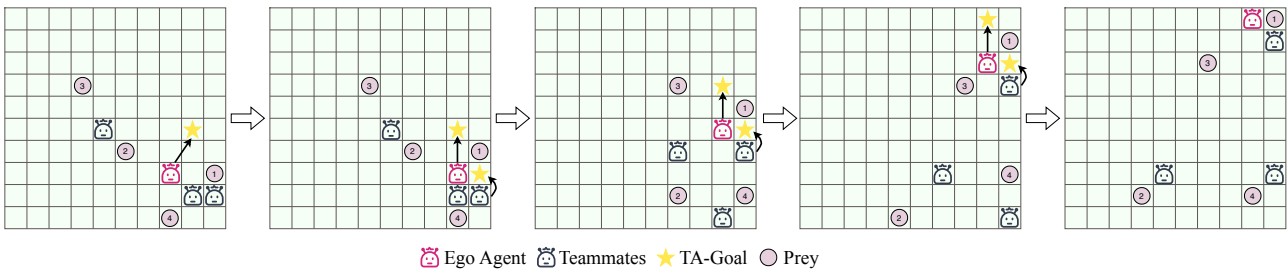

Figure 4. The visualization of one episode of the PP game, illustrates how the TA-Goal guides the ego agent (Red) to collaborate with its teammates in capturing the prey. At each step, instead of directly leading the ego agent to the prey, the TA-Goal leads the ego agent towards the optimal path with consideration and prediction of teammates' moves. The TA-Goal continuously adjusts as the agents move, ensuring that the ego agent responds to its teammates optimally.

replacing online interaction and buffer-based learning with direct learning from the offline dataset.

- **LIAM-off** (Papoudakis et al., 2021) is another offline-variant of online AHT method. To adapt LIAM into LIAM-off, we modify it to learn how to reconstruct global information from local observations directly from an offline dataset instead of a replay buffer, enabling the extraction of strong representational latent variables that guide the controlled agent's actions without requiring online interactions.

- **Conservative Q-Learning (CQL)** (Kumar et al., 2020) is a widely used offline RL algorithm that effectively addresses the overestimation issue by learning conser-

vative value functions. It penalizes unseen actions in the Q-function optimization, thus ensuring safer policy learning from static datasets. We adapt CQL to the AHT settings by training the ego agent in the offline data collected under diverse teammate behaviors.

- **Multi-Agent Decision Transformer (MADT)** (Meng et al., 2023) extends the Decision Transformer framework to multi-agent scenarios. It models the joint trajectory of all agents and leverages inter-agent dependencies using a unified transformer-based policy. We adapt it to the AHT settings, providing a strong baseline for evaluating offline AHT performance in settings where coordination and adaptation are essential.

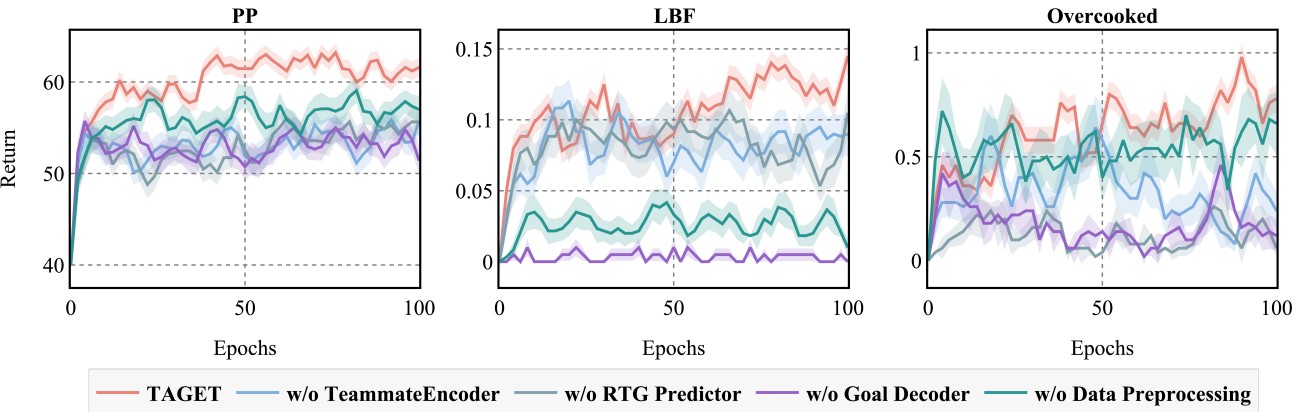

*Figure 5.* Impact of Model Components on Performance

## 5.3. Comparisons with Baselines

**Results**. We evaluate our trained model by interacting with teammates in the test teammates sets. These teammates are not involved in the collection of the offline training dataset, making this evaluation an effective measure of the model's generalization ability. Performance across different sizes of test teammate sets is evaluated for three environments. In the case of PP-8, one teammate set was randomly selected from eight distinct teammate sets to test our method in the PP environment, one teammate set represents a group of agents that interact with our ego agent in one episode, and similarly for other cases. The returns from each evaluation are averaged over fifty episodes of interaction. The shaded region represents the 95% confidence interval. The results, summarized in Appendix B, show that our method achieves superior performance across all testing environments.

**Analysis**. The results in Figure 3 provide a comprehensive comparison of five evaluated methods: **TAGET**, **DT**, **Prompt-DT**, **ODITS**, and **LIAM**. Returns are averaged over 50 trials, with shaded regions representing 95% confidence intervals using the standard normal distribution formula: $\bar{x} = \pm 1.96 \cdot \frac{\sigma}{\sqrt{n}}$, where $\sigma$ is the sample standard deviation and $n = 50$. TAGET consistently achieves the highest returns and demonstrates superior stability across all environments and test teammate set sizes, underscoring its strong generalization ability to unseen teammates, as evidenced by its higher curves. The consistent gap between TAGET and baselines such as DT and Prompt-DT demonstrates the ability of TAGET to effectively model diverse teammate behaviors, a critical challenge in AHT. DT and Prompt-DT show limited performance, with returns plateauing at lower levels. They lack explicit mechanisms to adapt to dynamically changing teammates in ad hoc scenarios. Prompt-DT, in particular, shows limited performance in environments where teammate behaviors change rapidly, as it relies on static prompts for guidance. In contrast, our

method introduces sub-goals as tokens that dynamically guide the agent's actions. These **TA-Goals**, predicted based on dynamic team context and TA-RTG, enable the ego agent to adapt to real-time changes in teammate behaviors effectively. Figure 4 shows a real trajectory sampled from the PP game environment, where the yellow pentagram represents the TA-Goal. It demonstrates how the TA-Goal guides the ego agent in cooperating with teammates and capturing the prey effectively. ODITS and LIAM occasionally achieve moderate returns but are characterized by higher variability and instability, particularly in larger test teammate sets (e.g., PP-8 and Overcooked-8). LIAM exhibits notable fluctuations, likely due to overfitting to specific teammate behaviors during training.

In summary, TAGET outperforms all baselines in terms of return, stability, and generalization across a wide range of multi-agent environments. This indicates that TAGET is better equipped to handle the complexities of cooperative tasks with diverse and unseen teammates, addressing key limitations observed in the baseline methods.

## 5.4. Ablation Study

To evaluate the contribution of each component in our proposed method, we conducted an ablation study by systematically removing individual components and observing the impact on performance across three benchmark environments. The results are summarized in Figure 5.

Across all environments, the full method (**TAGET**) consistently outperformed its ablated variants, demonstrating the importance of each component. Notably, the removal of the TA-Goal decoder led to the most significant performance drop, especially in the overcooked environment, where goal-driven actions are crucial due to poor teammate strategy. The absence of data pre-processing also led to consistent performance degradation in all three environments, underscoring its fundamental contribution to the model's generaliza-

tion ability. The impact was particularly evident in the LBF environment. By generating diverse interaction trajectories, our trajectory mirroring strategy effectively mitigates overfitting, enhancing the adaptability to unknown teammates and the effectiveness of offline learning. These findings strongly validate the design choices in our method, demonstrating that each component of our method contributes synergistically to its overall performance. This collective contribution significantly improves its adaptability in AHT scenarios.

## 6. Conclusion

In this work, we introduced TAGET, a novel hierarchical framework that effectively addresses the challenging problem of offline ad hoc teamwork. Through extensive experiments, we demonstrated that TAGET successfully overcomes three critical challenges: limited offline data through our mirrored data pre-processing strategy, partial observability via regularization of two state encoders, and online adaptation through teammate-aware return-to-go prediction. The strong empirical results across diverse cooperative environments validate the effectiveness of our hierarchical approach combining teammate-aware goal prediction with goal-conditioned decision making.

## Acknowledgements

This research is supported by Guangdong Basic and Applied Basic Research Foundation (2025A1515010247) and the Fundamental Research Funds for the Central Universities (2024ZYGXZR069).

## Impact Statement

Our work seeks to advance the field of multi-agent reinforcement learning by addressing the challenges of offline ad hoc teamwork with the proposed TAGET framework. We open up exciting opportunities for generative decision-making under the uncertainty of teammates, particularly in scenarios where agents must adapt to dynamic and unpredictable changes of teammates. Future research may consider more uncertainties from both the teammates and the environments, making the decision models more robust. We believe that our work facilitates the application of generative decision models in real-world uncertain scenarios.

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

# A. Details of Environments.

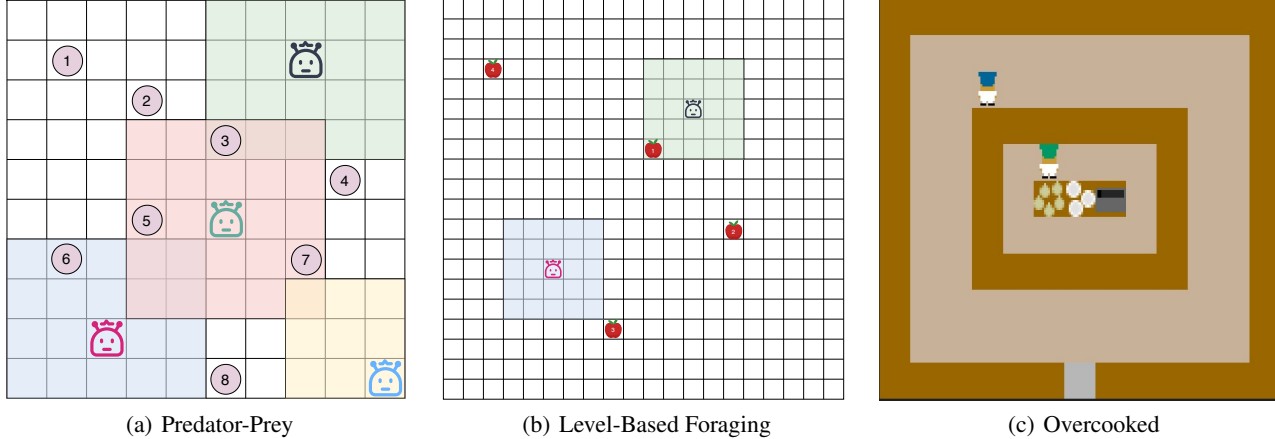

(a) Predator-Prey     (b) Level-Based Foraging     (c) Overcooked

*Figure 6.* Impact of different goal steps

**Level-based foraging.** In the Level-based Foraging (LBF) environment, two agents forage within a $20 \times 20$ grid world containing four food items. Each agent perceives the environment through a local observation space defined by a $5 \times 5$ window centered on its current position. The action space for each agent includes five discrete actions: four directional movements (up, down, left, right) and a "collect" action. The agents share a team reward, which is proportional to the level of the collected food and normalized to 1. An episode ends either when all food items are collected or after 50 timesteps. To enhance the challenge, a strict coordination rule is imposed: food can only be collected when both agents are adjacent to the same food item and simultaneously perform the "collect" action. This rule forces agents to synchronize both spatially and temporally, making cooperation essential. During evaluation, TAGET or baseline methods control one agent, while the other agent's behavior is governed by teammates sampled from predefined policy sets. The variability in coordination strategies is reflected in the sequence in which agents collect all food items.

**Predatory prey.** The Predator-Prey (PP) environment is a more complex variant of the LBF. In this scenario, two predator agents operate within a $10 \times 10$ grid world, each with a local $5 \times 5$ observation space centered on its current position. The objective is for the predators to capture four prey, which move randomly across the grid throughout the episode. The action space for each predator consists of four directional movement actions (up, down, left, and right). An episode ends either when all prey have been captured or after 200 timesteps. The random movement of the prey increases the difficulty of the task, as predators must maintain continuous coordination to effectively chase and surround them. To simplify the interaction, explicit "capture" actions are removed, and prey are considered captured once they are fully surrounded by both predators. This eliminates the risk of failed captures but still requires precise spatiotemporal coordination. The coordination strategies adopted by the predators are reflected in how they organize their pursuit and divide roles, as well as in the sequence in which prey are captured. This design highlights the importance of cooperative behaviors in multi-agent environments.

**Overcooked.** In the Overcooked environment, two agents must collaborate in a virtual kitchen to complete cooking tasks. Preparing a dish requires a series of actions and includes a waiting period. Once the dish is cooked, an agent must pick it up, transport it to the designated delivery point, and place it to complete an order. In this environment, both agents share an action space consisting of six discrete actions: moving in four directions, interacting with the object in front, and remaining idle. The objective is to complete as many orders as possible within 400 timesteps through efficient collaboration. Specifically, in our environment, we adopted the "Surrounding" layout, as shown in Fig. In this layout, only the green agent can operate the cookware, while only the blue agent can reach the delivery point. Since a single agent cannot independently complete the entire task, the two agents must cooperate closely to fulfill the orders. In this layout, coordination is mandatory, and agents must adapt to each other's behavior and preferences to complete the tasks.

## B. Experiment Results

| Methods | PP-4 | LBF-4 | Overcooked-4 | PP-8 | LBF-8 | Overcooked-8 |
|---|---|---|---|---|---|---|
| DT | $54.6 \pm 0.6$ | $0.098 \pm 0.008$ | $0.16 \pm 0.19$ | $55.4 \pm 1.6$ | $0.055 \pm 0.009$ | $0.08 \pm 0.03$ |
| Prompt-DT | $56.3 \pm 0.7$ | $0.102 \pm 0.010$ | $0.40 \pm 0.19$ | $54.4 \pm 1.2$ | $0.045 \pm 0.008$ | $0.64 \pm 0.15$ |
| ODITS-off | $55.6 \pm 1.6$ | $0.065 \pm 0.036$ | $0.24 \pm 0.28$ | $54.7 \pm 1.4$ | $0.000 \pm 0.000$ | $0.06 \pm 0.03$ |
| LIAM-off | $58.2 \pm 1.8$ | $0.005 \pm 0.009$ | $0.42 \pm 0.26$ | $55.3 \pm 1.8$ | $0.040 \pm 0.010$ | $0.24 \pm 0.09$ |
| CQL | $60.6 \pm 1.0$ | $0.065 \pm 0.036$ | $0.46 \pm 0.08$ | $59.0 \pm 0.8$ | $0.005 \pm 0.009$ | $0.36 \pm 0.08$ |
| MADT | $61.4 \pm 1.1$ | $0.010 \pm 0.008$ | $0.12 \pm 0.09$ | $62.4 \pm 0.6$ | $0.025 \pm 0.008$ | $0.12 \pm 0.09$ |
| **TAGET** | $\mathbf{63.2 \pm 1.1}$ | $\mathbf{0.140 \pm 0.080}$ | $\mathbf{0.98 \pm 0.15}$ | $\mathbf{62.9 \pm 1.3}$ | $\mathbf{0.080 \pm 0.010}$ | $\mathbf{0.77 \pm 0.15}$ |
| | +2.93% | +37.25% | +113.04% | +8.01% | +45.45% | +20.31% |

*Table 1.* Average Return Comparison with Baselines

## C. Impact of Goal Steps

The hyper-parameter $goal\_steps$ plays a crucial role in our algorithm, as it determines how far into the future the teammate observations are considered when generating sub-goals. Specifically, we define $goal\_steps$ as the number of steps ahead whose teammate observations are concatenated to form the sub-goal. These sub-goals are then used for supervised learning, guiding the agent to align its actions with future team objectives.

We evaluate the impact of different $goal\_steps$ by testing the values of 1, 2, 3, and 6 across three environments, the results are shown in Table 2. We observe that the optimal value of $goal\_steps$ depends on the complexity and dynamics of the environment. In Predator-Prey, the environment's simplicity and need for long-term planning result in the best performance with $goal\_steps$ = 6. A larger $goal\_steps$ enables the agent to make farsighted decisions, coordinating with teammates over extended horizons. In Level-Based Foraging, $goal\_steps$ = 2 achieves optimal results. In Overcooked, a highly complex environment, $goal\_steps$ = 3 performs best. Larger $goal\_steps$ such as 6 lead to reduced accuracy in sub-goal prediction, as the rapidly changing environment introduces noise and mismatches between predicted and actual teammate behaviors.

| | PP | LBF | Overcooked |
|---|---|---|---|
| Step=1 | $61.4 \pm 0.8$ | $0.145 \pm 0.009$ | $0.42 \pm 0.04$ |
| Step=2 | $60.1 \pm 0.7$ | $\mathbf{0.153 \pm 0.010}$ | $0.90 \pm 0.07$ |
| Step=3 | $59.6 \pm 0.7$ | $0.137 \pm 0.009$ | $\mathbf{0.98 \pm 0.15}$ |
| Step=6 | $\mathbf{63.2 \pm 1.1}$ | $0.123 \pm 0.008$ | $0.30 \pm 0.03$ |

*Table 2.* Average Return Comparison between different goal steps

## D. Pseudocode of Algorithm

Algorithm 1 demonstrates our trajectory mirroring strategy for pre-processing the offline dataset. This process iterates through all trajectories, generating new trajectories and adding them to the new dataset.

---
**Algorithm 1** Offline Data Pre-processing
---
**Input:** Pre-collected dataset $\mathcal{D} = \{\tau_1, \tau_2, \ldots, \tau_m\}$
**Output:** Offline dataset $\mathcal{D}'$
Initialize $\mathcal{D}' \leftarrow \emptyset$
**for** $j = 1$ **to** $m$ **do**
  **for** $i = 1$ **to** $n$ **do**
    Apply trajectory mirroring strategy to generate trajectory $\tau_j$:
    $\tau_{j,i} = f_{mirror}(\tau_j, i)$
    Add $\tau_{j,i}$ to $\mathcal{D}'$: $\mathcal{D}' \leftarrow \mathcal{D}' \cup \{\tau_{j,i}\}$
  **end for**
**end for**

---

Algorithm 2 demonstrates the offline training process of TAGET. It involves sampling trajectory batches from the offline dataset, computing team context and TA-RTG, generating TA-Goal and action predictions, and optimizing both high-level and low-level objectives using gradient descent. The training is repeated until the high-level and low-level losses converge.

---

**Algorithm 2** Offline training of TAGET

---

**Require:** Offline dataset $\mathcal{D}'$
**repeat**
    Sample a batch of trajectories $\mathcal{B} \subseteq \mathcal{D}'$
    **for** each trajectory $\tau \in \mathcal{B}$ **do**
        Randomly select a sub-sequence of length $3K$
        $L_{\text{High-level}} \leftarrow 0, L_{\text{DT}} \leftarrow 0$
        **for** each time step $t$ in sub-sequence **do**
            Compute $(\mu_{z_i}, \sigma_{z_i}) = f_\theta^*(o_t^i)$ , Sample $z_t^i \sim \mathcal{N}(\mu_{z_i}, \sigma_{z_i})$
            Compute $(\mu_{h_i}, \sigma_{h_i}) = f_\varphi(o_t^i, o_t^{-i})$ , Sample $h_t^i \sim \mathcal{N}(\mu_{h_i}, \sigma_{h_i})$
            Compute TA-RTG: $R_t = R_\phi(h_t^i)$
            Compute TA-Goal: $G_t = G_\xi(h_t^i, R_t)$
            Predict action $a_t^i$ using the low-level network
            Compute $L_{Reg}, L_{RTG}, L_{TA-Goal}, L_{DT}$ using Eq.6, Eq.7, Eq.8, Eq.10
            Accumulate high-level loss $L_{\text{High-level}}$ using Eq. 9
            Accumulate low-level loss $L_{\text{DT}}$ using Eq. 10
        **end for**
        Update parameters $\psi, \varphi, \theta, \phi, \xi$ using gradient descent
    **end for**
**until** convergence of high-level loss $L_{\text{High-level}}$ and low-level loss $L_{\text{DT}}$

---

Algorithm 3 illustrates the online testing process of TAGET. It updates the sequence iteratively using TA-Goal, observations, and actions. Based on the updated sequence, the low-level network generates action decisions, enabling adaptive and context-aware decision-making throughout the testing phase.

---

**Algorithm 3** Online testing of TAGET

---

**Require:** parameters $\psi, \theta, \phi, \xi$
Initialize sequence $S \leftarrow \emptyset$
**for** t = 1, ..., T **do**
    Keep the length of the sequence $S$ as $3K$
    Compute $(\mu_{z_i}, \sigma_{z_i}) = f_\theta^*(o_t^i)$ , Sample Team Context $z_t^i \sim \mathcal{N}(\mu_{z_i}, \sigma_{z_i})$
    Compute TA-RTG: $\hat{R}_t = R_\phi(z_t^i)$
    Compute TA-Goal: $\hat{G}_t = G_\xi(z_t^i, \hat{R}_t)$
    Add $\{\hat{G}_t, o_t^i\}$ into sequence $S$
    Generate the action $a_t^i$ through low-level network using $S$
    Add $\{a_t^i\}$ into sequence $S$
**end for**

---

# E. Network Architecture and Implementation Details

We illustrate the overall network structure of our method in Figure 7. Our model adopts the Decision Transformer (DT) backbone, with the following configurations: an embedding dimension of 64, context window length $K = 30$, 2 transformer layers with 1 attention head each, ReLU activation, and a dropout rate of 0.3. The network is optimized using AdamW with a learning rate of 0.01, batch size of 2048, and a weight decay of 0.0001. There are several task-specific coefficients to balance different learning objectives in our training loss. Specifically, we set the weighting parameters as follows: $\alpha = 0.0001$, $\beta = 100$, $\gamma = 100$, and $\sigma = 0.001$

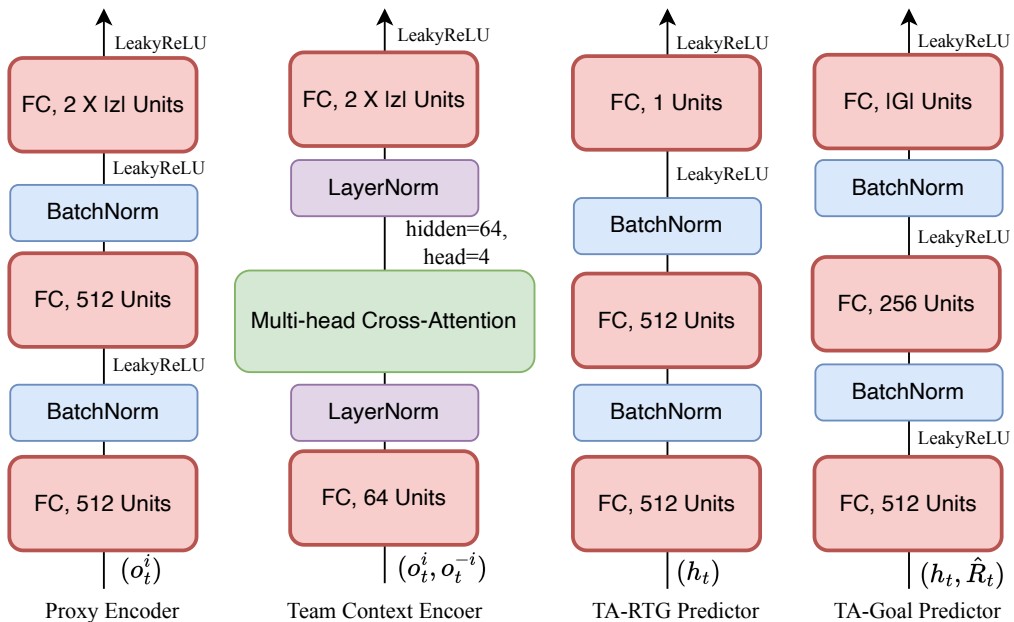

*Figure 7.* Architectural details of TAGET

