# OpenReview forum: "Ad Hoc Teamwork via Offline Goal-Based Decision Transformers"
_ICML.cc/2025/Conference — ICML 2025 poster_

### Official Review · Reviewer_nbn1 · 2025-02-19

**Overall Recommendation:** 4

**Summary:**

The paper introduces TAGET, a hierarchical framework for offline ad hoc teamwork (AHT) that leverages teammate-aware goal-conditioned Decision Transformers. The core idea is to dynamically predict teammate-aware return-to-go (TA-RTG) and sub-goals (TA-Goals) to enable real-time adaptation to unknown teammates without online interactions. Key innovations include a trajectory mirroring strategy for data efficiency, dual state encoders with regularization for partial observability, and a hierarchical structure combining high-level goal prediction with low-level action generation. Experiments in three environments demonstrate TAGET’s superiority over baselines, with an average performance improvement of 48.08%. Ablation studies validate the contributions of individual components, such as TA-Goal decoding and data preprocessing.

**Claims And Evidence:**

The authors raise issues such as poor generalization of the learned policy and challenges in accurately modeling teammates. However, the subsequent experimental section does not clearly demonstrate whether the proposed method effectively addresses these problems. For instance, how accurate is the modeling of teammates, and how well does the policy generalize?

**Essential References Not Discussed:**

[1] **multi-agent decision transformer**: Meng, Linghui, et al. "Offline pre-trained multi-agent decision transformer: One big sequence model tackles all smac tasks." arXiv preprint arXiv:2112.02845 (2021).

[2] **offline teammate modeling**: Zhu, Zhengbang, et al. "Madiff: Offline multi-agent learning with diffusion models." arXiv preprint arXiv:2305.17330 (2023).

**Experimental Designs Or Analyses:**

The evaluation of teammate modeling accuracy, policy generalization, and the ablation of return in the return-to-go function is missing. Additionally, experiments involving humans as teammates are not included. The experimental scenarios are relatively simple, and further testing in more complex environments, such as SMAC [1], is needed.

[1] Wang, Caroline, et al. "N-Agent Ad Hoc Teamwork." arXiv preprint arXiv:2404.10740 (2024).

**Methods And Evaluation Criteria:**

The method and validation don’t quite make sense. In my past experience with offline reinforcement learning, data augmentation doesn’t provide significant improvements in offline scenarios. Moreover, the accuracy of Teammate-aware goal prediction is hard to guarantee, and no experimental validation is provided.

**Other Comments Or Suggestions:**

No.

**Other Strengths And Weaknesses:**

**Strengths**:

1. The offline AHT problem addressed is novel.

**Weaknesses**:

1. The method lacks innovation. Techniques such as dataset augmentation, local observation-based global state prediction, and decision transformers are not new. The authors need to further clarify how these techniques are organically combined.

2. The experimental section is insufficient, lacking evaluations of teammate modeling accuracy, policy generalization, and ablation studies on the return in the return-to-go function.

3. The method still relies on algorithms that collect data driven by diversity. I’m curious about the method’s robustness when applied to low-diversity datasets.

**Questions For Authors:**

1. What is the difference between teammate-aware return-to-go and conventional return-to-go? (Has it shifted from observation to global state?)

2. How does the method address the issue of “poor generalization of the learned policy” mentioned in the paper? If this isn’t addressed, it remains a point of concern.

3. Can the trajectory mirroring strategy be understood as a form of data augmentation? How is it different from traditional data augmentation? How does this strategy work when the dataset’s diversity is low?

4. What is the impact of dataset quality on performance? Does a dataset with low diversity greatly affect the performance of the method?

5. Can you further clarify how the various components of the method are integrated?

**Relation To Broader Scientific Literature:**

This paper clearly situates TAGET within offline RL (e.g., Decision Transformer) and AHT (e.g., ODITS, LIAM), and highlights gaps in prior AHT methods (online dependency) and offline RL (lack of teammate adaptability).

**Theoretical Claims:**

No, there are no proofs in this paper.

---

> ### Author Rebuttal · Authors · 2025-03-31
>
> **Q1.  Questions in Claims And Evidence.**
>
> A1.  We apologize for any confusion. Firstly, we perform implicit modeling of teammates to capture their changes and update our TA-goal, so there is no explicit metric to measure the accuracy of teammate modeling. As for the generalization of the policy, we demonstrate this in our experiments. Specifically, we test the policy with teammates that have never been seen during training. The fact that our method achieves higher scores when cooperating with these unknown teammates, compared to other baseline methods, serves as evidence of the generalization ability of our proposed approach.
>
> **Q2. Questions in Methods And Evaluation Criteria.**
>
> A2. Regarding your concerns about data augmentation, more details can be found in our response to Reviewer TMEP, Response A1. Regarding the accuracy of TA-Goal, during training, we use the real state after $k$ steps as the ground truth, which is a common practice in many works. During testing, since there are no trajectories for reference, we cannot directly measure the accuracy of TA-goal's prediction. Our goal during testing is not to achieve the absolute accuracy of the TA-Goal but to assess its ability to generalize to unknown teammates in AHT.
>
> **Q3. Questions in Experimental Designs Or Analyses.**
>
> A3. The evaluation of teammate modeling accuracy and policy generalization is answered in A1. Since TA-RTG is a predicted value rather than a fixed input, it cannot be ablated in the traditional sense. Besides, we agree that further testing in more complex environments, such as SMAC, is necessary, and we plan to conduct such tests.
>
> **Q4. Questions in Essential References Not Discussed.**
>
> A4. Thank you for pointing out these relevant works. We acknowledge the importance of the two and will include a discussion of them in the revised manuscript to better situate our contributions within the existing literature.
>
> **Q5. Concern about innovation.**
>
> A5. Thank you for the comment. Our method is not designed to simply combine existing techniques, but is driven by a novel problem setting - offline AHT. This new setting raises unique challenges, which motivate our methodological choices. Rather than applying existing tools directly, we analyze why previous approaches like DT are insufficient and propose tailored solutions.
>
> Our trajectory mirroring strategy is a data preprocessing method specifically designed for the AHT setting. It improves data efficiency and teammate diversity without introducing extra computational cost. After preprocessing, we learn policies from offline data. While we adopt the Decision Transformer (DT) architecture, we find that standard DT is not well-suited for our problem, as RTG cannot capture dynamic teammate behavior. To address this, we propose TA-RTG, a novel formulation that incorporates teammate-awareness. Furthermore, given the dynamic nature of the environment, we propose the novel concept of TA-Goal, a sub-goal representation predicted from TA-RTG, to guide the transformer’s decision-making in a hierarchical manner.
>
> **Q6. Concern about the method’s robustness when applied to low-diversity datasets.**
>
> A6. Thank you for raising this important point. While our method benefits from diverse offline data, we also aim to improve robustness through model design. Specifically, the proposed TA-RTG and TA-Goal mechanisms help the model infer and adapt to teammate behavior even under limited diversity. In our ablation studies, the variant without the trajectory mirroring strategy can be seen as a setting with reduced data diversity. We observe that the method still maintains reasonable performance under this condition, demonstrating a degree of robustness to limited diversity. Nonetheless, we acknowledge this as a valuable direction and plan to further explore robustness under low-diversity settings in future work.
>
> **Q7. Difference between TA-RTG and RTG.**
>
> A7.  The main difference between the two is that TA-RTG considers teammate behavior in a team-shared reward setting. TA-RTG incorporates how teammates' actions influence the collective reward, adapting the agent's strategy based on the team context. TA-RTG shifts from relying solely on the agent’s observation to considering the team context, allowing it to adapt to changes in teammate behavior (TA-RTG and RTG ablation: https://anonymous.4open.science/api/repo/materials-E410/file/ablation_new.pdf).
>
> **Q8. Concern about poor generalization.**
>
> A8. Thank you for highlighting this important point. Our method addresses the generalization issue through both data-level and model-level designs. On the data side, the trajectory mirroring strategy increases teammate diversity. On the model side, the proposed TA-RTG and TA-Goal explicitly encode teammate behavior and adapt decision-making accordingly. These components together improve the model’s ability to generalize to unseen teammates. We will clarify this contribution further in the revised manuscript.

---

### Official Review · Reviewer_85vr · 2025-03-09

**Overall Recommendation:** 2

**Summary:**

The paper presents TAGET, a novel hierarchical framework designed to address the challenges of ad hoc teamwork in settings where only offline multi-agent interaction data is available. Traditional ad hoc teamwork approaches rely on real-time interactions, but TAGET circumvents the need for costly online simulations by learning robust policies from pre-collected datasets. At its core, TAGET integrates a high-level module that dynamically predicts a teammate-aware sub-goal, leveraging a new teammate-aware return-to-go signal and dual state encoders to capture the evolving team contex, with a low-level, goal-conditioned Decision Transformer that generates adaptive actions. The framework also incorporates a trajectory mirroring strategy that repurposes each multi-agent trajectory by cyclically treating each agent as the ego agent, thereby enhancing data efficiency and capturing diverse decision-making perspectives. Extensive experiments conducted in environments such as Predator-Prey, Level-Based Foraging, and Overcooked demonstrate that TAGET significantly outperforms existing baselines in terms of overall return, stability, and generalization to unseen teammate behaviors, marking a substantial advancement in offline multi-agent reinforcement learning.

## Update After Rebuttal
The authors have addressed part of my concerns. However, I still have some concerns about its potential effect: I have a different view of the feasibility of transformer based method is the route towards the final goal of juggling realistic deployment and offline training. The features of decentralized execution and parallel decision making is crucial to my best experience.

For this reason, I raised my score from 1 to 2.

**Claims And Evidence:**

### Supported Claims:
1. The claims of better performance than other offline methods are supported by strong experimental evidence.

### Problematic Claims:
1. The authors claim that "TA-Goal captures the dynamics of teammate behaviors, providing more robust and adaptive guidance than RTG in complex multi-agent environments." However, I didn't see the evidence to support this claim and it is non-trivial to understand this claim by common sense.
2. The authors claim that "To model this, we represent the team context in a stochastic embedding space $\mathcal{H}$, where it is encoded as a latent probabilistic variable $h_{i}$ drawn from a multivariate Gaussian distribution." However, I didn't see any evidence to support this viewpoint. Why is the team context modeled as a multivariate random variable? Furthermore, why is it Gaussian distribution?
3. The authors claim that "Similarly, $z_{i}$, as an approximation of $h_{i}$, is encoded into a stochastic embedding space, where it is represented as a latent probabilistic variable drawn from a multivariate Gaussian distribution." Can the authors provide any evidence to support this modeling strategy?
4. It seems like in Eq. (6), the first KL-divergence is a reverse KL-divergence. Can the authors provide more insight into this design?
5. In Eq. (8), the groundtruth goal is constructed by the concatenation of team observations. As a result, it is not a binary variable. Can the authors give an explanation about why the loss in Eq. (8) is formulated as binary cross entropy?

**Essential References Not Discussed:**

Another recently published paper in ICML 2024 [1] which studies online interactions for AHT could be cited in related works to enhance the understanding of the main contribution of this paper.

[1] Wang J, Li Y, Zhang Y, Pan W, Kaski S. Open Ad Hoc Teamwork with Cooperative Game Theory. In International Conference on Machine Learning 2024 Jul 8 (pp. 50902-50930). PMLR.

**Experimental Designs Or Analyses:**

In general, the experimental design is following the conventional experimental setup of AHT.

However, how the teammate sets are generated may miss some information. The authors have mentioned that they have trained four distinct populations of MARL policies. What are the MARL algorithms used? How are the four distinct populations formed?

**Methods And Evaluation Criteria:**

### Evaluation Criteria:
The evaluation criateria used in this paper is acceptable to the AHT problem.

### Methods:
The authors claim that the offline data is easy to collect and propose to use decision transformer as an offline RL method to address AHT. However, I have some concerns on the difficulty of collecting offline data, which may affect the reasonableness of the proposed method. First, if we consider collecting offline samples from simulators, why not train the ego agent through interaction with environment? On the other hand, if we consider collecting data from real-world scenarios, what is the way to capture long-horizon rewards? It is not easy to track a team of agents constantly in a long episode, especially with accurate records of data. For example, the traffic data captured by CCTV cameras is with lower frequency than the control rate by an agent. Let's say that the distance between two cameras is 1 km, and there may happen multiple decision making steps. On the other hand, if each agent is able to collect data by the camera installed on its body, the alignment between timestamps could extremely difficult. In conclusion, offline data collection is not as easy as what the authors claimed.

**Other Comments Or Suggestions:**

In Eq. (8), $\hat{G}$ and $G$ interchangebly represent the groundtruth in multiple places. I believe this could be a typo.

**Other Strengths And Weaknesses:**

1. The methods are well described, however, the reason behind part of the procedure is not explained.
2. The proposed algorithm is novel with an idea to use offline collected data, though the applicability to real-world problems is still under concern.

**Questions For Authors:**

The authors are required to make clarification for the problematic claims listed in **Claims And Evidence**. In addition, the authors are requested to answer the concerns about the applicability of methods in **Methods And Evaluation Criteria** and the dataset generation in **Experimental Designs Or Analyses**.

**Relation To Broader Scientific Literature:**

In prior work, the ego agent (learner) was not trained by the offline dataset. As I mentioned above, this could be due to the validity and difficulty of collecting offline data. In turn, this could lead to a question that if offline RL paradigm does fit in the aim of the AHT problem.

The major issue of previous methods is how to construct diverse teammate sets (models). With the same techniques of establishing diverse teammate sets, I suspect if the offline method proposed in this paper is necessary. The direct evidence is that in experiments it seems like the proposed offline method still requires diverse teammate sets, like what is required in AHT with online interaction.

If the problem of offline data collection is dismissed, the major contribution of this paper is proposing a training paradigm which does not need online interactions. The remaining questions is that if the offline collected data without interactions can actually have good quality as the data collected with interactions.

**Theoretical Claims:**

This paper has no theoretical claims.

---

> ### Author Rebuttal · Authors · 2025-03-31
>
> Thank you for your suggestions.
>
> **Q1. Concerns on the difficulty of collecting offline data and the applications of methods**
>
> A1.We apologize for the misunderstanding. Our claim that offline data is "relatively easy to collect" refers to its cost and safety advantages over online training, not to ignore practical challenges. We use simulator-collected data to avoid exploration risks (e.g., collisions in autonomous driving) and improve data efficiency with trajectory mirroring. For real-world data, we address challenges like long-horizon rewards using multi-sensor fusion [1] and hierarchical reward decomposition [2]. While data collection is challenging, offline RL remains valuable for providing a safe, low-cost baseline and enabling future online refinement.
>
> **Q2. Concerns on empirical validation and intuitive explanation of TA-Goal's advantages over RTG**
>
> A2.Towards empirical validation, our experiments (Fig. 3) show that TAGET (TA-Goal-based) outperforms DT (RTG-based). Intuitively,  RTG is a long-term scalar target that doesn't adapt to changing teammate behaviors (e.g., obstacle avoidance). TA-Goal, on the other hand, dynamically adjusts based on teammate intent, enabling adaptive actions (e.g., flanking maneuvers) as shown in Fig. 4. https://anonymous.4open.science/api/repo/materials-E410/file/ablation_new.pdf
>
> **Q3. Why is the team context modeled as a multivariate random variable and selected Gaussian distribution**
>
> A3. The team context is modeled as a multivariate random variable to capture uncertainties in teammates' policies, enhancing decision robustness. The Gaussian distribution is chosen due to: (1) mathematical and computational advantages; (2) uncertainty modeling capability (3) research alignment—Gaussians are standard in deep probabilistic models (e.g., VAEs [3][4] ). Many existing works [5] adopt this approach, and we will clarify this in our paper more clearly.
>
> **Q4. The evidence to support this modeling strategy.**
>
> A4.  In partially observable settings, the ego agent infers the global team context $h_i$ from local observations $o_i$. KL-divergence regularization (Eq. 6) aligns $z_i$ with $h_i$, enabling latent team dynamics capture without global info. This method has been validated in previous works [5][6] for improved generalization. If additional evidence is needed, we will include it in the revised manuscript.
>
> **Q5. Eq.(6). The first KL-divergence is a reverse KL-divergence.**
>
> A5. The first term in Eq. (6) is a reverse KL-divergence, following the beta-VAE approach [4], which encourages the latent representation to match a standard normal prior, preventing overfitting and improving generalization. The second term uses forward KL to align the proxy and team context representations.
>
> **Q6. Eq.(8) explanation about why the loss is formulated as binary cross entropy**
>
> A6. Thanks for the useful feedback. The groundtruth goal $\hat{G}$ is constructed by concatenating discretized observations (e.g., grid-world positions, task states) and one-hot encoding the global state $s_{t+k}$, where each dimension represents a binary indicator. We use BCE loss per dimension to handle binary classification, which aligns with the discrete nature of our environments. Using MSE for continuous regression would be suboptimal, as the goal representation is categorical and requires probabilistic modeling of binary outcomes. We will also fix the inconsistency between $\hat{G}$ and $G$.
>
> **Q7. Concerns about the applicability of methods in Methods And Evaluation Criteria.**
>
> A7. While collecting offline datasets is challenging, it is manageable in AHT. The core challenge is enabling the ego agent to handle unknown teammates, which previous works have addressed in two ways: achieving generalization through algorithms and constructing diverse sets of teammates. Even with algorithmic generalization, a diverse offline dataset is still needed for better performance, which doesn’t mean it’s not worth exploring. The offline method is not meant to replace online interactions but is motivated by safety and cost considerations, and can be further optimized when combined with online methods.
>
> **Q8. The dataset generation in Experimental Designs Or Analyses.**
>
> A8. Thank you for your feedback. For more details, please refer to our response to Reviewer cnPV, A1.
>
> * [1] Li, et al. "Dynamic Semantic SLAM Based on Panoramic Camera and LiDAR Fusion for Autonomous Driving." IEEE TAITS 2025.
> * [2] Matsuda, et al. "Hierarchical Reward Model of Deep Reinforcement Learning for Enhancing Cooperative Behavior in Automated Driving." JACII 2024.
> * [3] Kingma, et al. "Auto-encoding variational bayes." 20 Dec. 2013.
> * [4] Higgins, et al. "beta-vae: Learning basic visual concepts with a constrained variational framework." ICLR. 2017.
> * [5] Gu, et al. "Online ad hoc teamwork under partial observability." ICLR. 2021.
> * [6] Xie, et al. "Future-conditioned unsupervised pretraining for decision transformer." ICML, 2023.

---

> > ### Comment · Reviewer_85vr · 2025-04-02
> >
> > 1. The answer for Q1 is acceptable to me, but I hope the authors can complement the introduction with these faithful statements.
> >
> > 2. I suggest the authors complement the evidence to support the modeling strategy as well as Q5 and Q6 in the revised maniscript, to deliver a high-quality and responsible work. This will not only benefit the future work, but also prevent the risk of misleading others.
> >
> > 3. About the applicability of using transformer, I never judge its usefulness to tackle off-line training. However, I believe that the incorporation of benefit of tackling off-line training should be built upon some decision-making paradigms that are adaptable to existing recognized scenarios. Otherwise, it will highly likely lead to a result that some tachniques are developed independent of what was needed in priority in real world.

---

> > > ### Author Response · Authors · 2025-04-04
> > >
> > > Thanks for your insightful feedback.
> > >
> > > **Q1. Complement the introduction with these faithful statements.**
> > >
> > > A1. Thank you for your constructive feedback. We have revised the introduction (Lines 19–25) with the faithful statements. The updated text now reads:
> > >
> > > "Environmental simulators are often unavailable or expensive, especially for real-world scenarios such as autonomous driving or disaster response, where online RL approaches face prohibitive safety risks (e.g., collisions or mission failures during exploration). However, offline data collection **provides a safer and more cost-effective alternative**—multi-agent interaction datasets can be acquired through pre-recorded logs (e.g., traffic data from cameras and road sensors) or synthetic simulations. While challenges such as long-horizon reward assignment and partial observability persist, recent advances in offline RL (e.g., multi-sensor fusion [1] and hierarchical reward decomposition [2]) enable practical solutions. In this work, we further improve data efficiency via a trajectory mirroring strategy, amplifying the utility of limited datasets by treating each agent as the ego agent. These advantages make offline AHT a viable foundation for real-world deployment, offering a safe baseline policy that can later be refined through online adaptation."
> > >
> > > **Q2. Complement the evidence to support the modeling strategy as well as Q5 and Q6 in the revised manuscript.**
> > >
> > > A2.  We have revised the descriptions of  Eq.(6) (Lines 256–265) and Eq.(8) (Lines 254-255).
> > >
> > > "The first KL-divergence term $D_{KL}( f_{\varphi}(\cdot | o_t^i, o_t^{-i})$ is designed as a reverse KL to enforce a compact latent space. Following the beta-VAE framework [3], reverse KL encourages the learned latent distribution to concentrate around the modes of the prior (standard normal distribution), effectively preventing overfitting to noisy observations and improving generalization to unseen teammates. In contrast, the second term employs forward KL, which aligns the proxy encoder’s output with the team context encoder by minimizing the divergence in expectation. This hybrid regularization ensures the latent space regularity and the consistency between local and global representations under partial observability. "
> > >
> > > "The groundtruth goal $\hat{G}$ is constructed by concatenating discretized observations (e.g., grid-world positions, task states) and one-hot encoding the global state, where each dimension represents a binary indicator. The loss function is defined as: "
> > >
> > > **Q3. Transformer-based offline training techniques must align closely with established decision-making paradigms in real-world scenarios.**
> > >
> > > AHT captures a core challenge in real-world multi-agent systems: enabling an ego agent to collaborate with unknown teammates without prior coordination. Unlike centralized MARL frameworks that require synchronized training of all agents—a scenario rarely feasible in practice (e.g., autonomous cars cannot jointly train with all possible human drivers)—AHT directly models the ego-centric adaptation paradigm. **This aligns with real-world applications** where agents must operate in open environments with changing teammates.
> > >
> > > **The success of offline MARL** [4] demonstrates that effective policies can be learned without online interactions, making its extension to AHT (a specialized case of MARL) a natural and viable direction. Beyond theory, offline AHT offers **practical value**: in autonomous driving, vehicles learn adaptive strategies from pre-collected data, avoiding collision risks and real-world testing costs. Besides, in healthcare robotics, surgical assistants trained offline on historical operation logs can adapt to diverse surgeon workflows without risking patient safety during live procedures. These applications ensure efficiency and safety across critical domains.
> > >
> > > DT is uniquely suited for offline AHT due to its ability to model long-term dependencies in sequential decision-making. However, vanilla DT struggles in AHT due to its reliance on scalar return-to-go (RTG), which fails to capture dynamic team objectives. We propose the novel concept of TA-Goal, a sub-goal representation predicted from TA-RTG, to guide the transformer’s decision-making in a hierarchical manner.
> > >
> > > To our knowledge, this is the first work addressing offline AHT, which we hope will inspire future research and practical applications. We would be grateful if you could reevaluate our work.
> > >
> > > * [1] Li, et al. "Dynamic Semantic SLAM Based on Panoramic Camera and LiDAR Fusion for Autonomous Driving." IEEE TAITS 2025.
> > > * [2] Matsuda, et al. "Hierarchical Reward Model of Deep Reinforcement Learning for Enhancing Cooperative Behavior in Automated Driving." JACII 2024.
> > > * [3] Higgins, et al. "beta-vae: Learning basic visual concepts with a constrained variational framework." ICLR 2017.
> > > * [4] Yang, et al. "Believe what you see: Implicit constraint approach for offline multi-agent reinforcement learning." NeurIPS 2021.

---

### Official Review · Reviewer_TMEP · 2025-03-13

**Overall Recommendation:** 2

**Summary:**

This paper introduces **TAGET (Teammate-Aware Goal driven hiErarchical Decision Transformers)**, a novel framework for offline ad hoc teamwork (AHT). The AHT problem requires an agent (ego agent) to collaborate with unknown teammates without prior coordination. Unlike existing approaches that rely on online reinforcement learning (RL) and direct environmental interactions, TAGET enables offline learning from pre-collected multi-agent datasets, addressing critical challenges such as limited data availability, partial observability, and dynamic teammate adaptation.

**Claims And Evidence:**

1. The trajectory mirroring strategy is claimed to enhance data efficiency, and ablation results (Figure 5) support this. However, no alternative data augmentation methods are tested, making it unclear if mirroring is the best approach. Testing other augmentation techniques (e.g., data mixing or adversarial augmentation) would provide stronger evidence.


2. The paper assumes TAGET is robust to diverse teammate behaviors, but no systematic study on the impact of teammate diversity is provided. Evaluating performance across different levels of teammate heterogeneity would make the generalization claim more convincing.

**Essential References Not Discussed:**

The references in this paper are sufficient.

**Experimental Designs Or Analyses:**

The detailed experimental designs are not clarified, such as parameter selection, model architecture, encoder/decoder selection. Authors should include the main algorithm in the main paper, while putting all the substantial tricks into the appendix.

**Methods And Evaluation Criteria:**

1. The hierarchical TAGET framework aligns well with the offline ad hoc teamwork (AHT) problem, effectively addressing data scarcity, partial observability, and dynamic teammate adaptation. The trajectory mirroring strategy is a reasonable approach to improve dataset efficiency.

2. The evaluation on Predator-Prey, Level-Based Foraging, and Overcooked is appropriate for testing multi-agent coordination and adaptability.

3. Baseline comparisons include DT, Prompt-DT, ODITS-off, and LIAM-off, which are relevant, but classic offline RL methods like CQL [1] or QT [2] should be included for a more comprehensive benchmark suite.

4. Evaluation metrics focus on return scores, which effectively measure task performance. However, no explicit measure of adaptability to new teammates over time is provided. A metric tracking adaptation efficiency would strengthen the evaluation.

5. Authors did not explicitly claim how to compute standard errors with repeat experiments, which makes it hard to tell whether the improvements are meaningful. Computational time of each algorithm should also be listed for fair comparison.

Overall, the methods are well-designed, but the evaluation could be strengthened with real-world benchmarks, additional baselines, and statistical validation.

- [1] Kumar, A., Zhou, A., Tucker, G., & Levine, S. (2020). Conservative q-learning for offline reinforcement learning. Advances in neural information processing systems, 33, 1179-1191.

- [2] Kalashnikov, D., Irpan, A., Pastor, P., Ibarz, J., Herzog, A., Jang, E., ... & Levine, S. (2018, October). Scalable deep reinforcement learning for vision-based robotic manipulation. In Conference on robot learning (pp. 651-673). PMLR.

**Other Comments Or Suggestions:**

Please refer to the “Other Strengths And Weaknesses” section.

**Other Strengths And Weaknesses:**

## Strengths:
1. Novel Offline Ad Hoc Teamwork Framework – The paper introduces TAGET, a hierarchical decision-transformer-based approach to offline AHT, addressing critical challenges such as limited data, partial observability, and teammate adaptation.

2. Effective Use of Goal-Based Adaptation – The introduction of Teammate-Aware Goal (TA-Goal) and Teammate-Aware Return-to-Go (TA-RTG) is a significant innovation that helps the ego agent dynamically adapt to unseen teammates without relying on explicit online interactions.

3. Data Efficiency via Trajectory Mirroring – The trajectory mirroring strategy improves the use of offline datasets by allowing each agent to be treated as an ego agent in different contexts, effectively increasing the amount of usable training data.

4. Addresses a Practical Limitation of Online AHT – Most prior ad hoc teamwork approaches require online interactions, which are impractical in real-world applications. TAGET’s offline approach makes AHT more feasible for real-world deployment.


## Weaknesses:

1. Limited Novelty in Offline RL Techniques – While TAGET innovates in offline AHT, its use of hierarchical decision transformers and return-to-go estimation builds on existing offline RL techniques (e.g., Decision Transformer, CQL) rather than introducing fundamentally new RL methods. A clearer differentiation from prior offline RL works would strengthen its contribution.

2. No Direct Ablation for TA-RTG Effectiveness – The paper claims that Teammate-Aware Return-to-Go (TA-RTG) is superior to standard RTG, but no explicit ablation study compares TAGET with standard return-to-go instead of TA-RTG. This omission makes it unclear whether TA-RTG is necessary for TAGET’s success.

3. Lack of Analysis on Teammate Diversity Impact – The generalization of TAGET to diverse teammates is assumed, but the paper does not systematically analyze how teammate heterogeneity affects performance. Testing against teammates with drastically different strategies or levels of expertise would provide a stronger understanding of its adaptability.

4. Confusing Figures and Lack of Detailed Descriptions – Some figures lack clear explanations (e.g., Figure 3 performance curves do not explicitly mention how the 95% confidence intervals are computed, and no verbal explanations of those curves are provided). The captions should give more precise descriptions of experimental setups and key observations.

**Questions For Authors:**

Please refer to the “Other Strengths And Weaknesses” section.

**Relation To Broader Scientific Literature:**

This paper introduces TAGET, addressing key limitations of prior ad hoc teamwork (AHT) methods in multi-agent reinforcement learning (MARL), particularly in offline settings. Previous online AHT methods rely heavily on environment interactions, limiting generalization and adaptability to unknown teammates. TAGET overcomes these issues by learning teammate-aware goal-driven policies from offline datasets.

Traditional online AHT approaches [1][2] require continuous adaptation to teammates through reinforcement learning (RL), which is infeasible without environmental access. TAGET builds on offline RL advancements (e.g., Decision Transformer [3], Conservative Q-Learning [4]), reformulating AHT as a sequence modeling problem while introducing TA-Goal and TA-RTG for better teammate modeling.
Unlike prior offline MARL works [5][6] that learn joint optimal policies, TAGET focuses on real-time adaptability to changing teammates. The trajectory mirroring strategy further differentiates TAGET by enhancing dataset efficiency, addressing data limitations in offline settings.
Additionally, prior teammate/opponent modeling methods [7][8] rely on explicit policy inference, which struggles in offline settings due to missing exploration opportunities. TAGET circumvents this by predicting future global states (TA-Goal) rather than teammates' policies, making it more robust to unseen agents.

By integrating ideas from offline RL, sequence modeling, and multi-agent adaptation, TAGET advances offline AHT research and sets a foundation for applying decision transformers in multi-agent collaboration without online interactions.

- [1] Barrett, S., & Stone, P. (2015, February). Cooperating with unknown teammates in complex domains: A robot soccer case study of ad hoc teamwork. In Proceedings of the AAAI Conference on Artificial Intelligence (Vol. 29, No. 1).

- [2] Rahman, M. A., Hopner, N., Christianos, F., & Albrecht, S. V. (2021, July). Towards open ad hoc teamwork using graph-based policy learning. In International conference on machine learning (pp. 8776-8786). PMLR.

- [3] Chen, L., Lu, K., Rajeswaran, A., Lee, K., Grover, A., Laskin, M., ... & Mordatch, I. (2021). Decision transformer: Reinforcement learning via sequence modeling. Advances in neural information processing systems, 34, 15084-15097.

- [4] Kumar, A., Zhou, A., Tucker, G., & Levine, S. (2020). Conservative q-learning for offline reinforcement learning. Advances in neural information processing systems, 33, 1179-1191.

- [5] Tseng, W. C., Wang, T. H. J., Lin, Y. C., & Isola, P. (2022). Offline multi-agent reinforcement learning with knowledge distillation. Advances in Neural Information Processing Systems, 35, 226-237.

- [6] Jiang, J., & Lu, Z. (2023). Offline Decentralized Multi-Agent Reinforcement Learning. In ECAI (pp. 1148-1155).

- [7] Gu, P., Zhao, M., Hao, J., & An, B. (2021). Online ad hoc teamwork under partial observability. In International conference on learning representations.

- [8] Zintgraf, L., Devlin, S., Ciosek, K., Whiteson, S., & Hofmann, K. (2021). Deep Interactive Bayesian Reinforcement Learning via Meta-Learning. In Proceedings of the 20th International Conference on Autonomous Agents and MultiAgent Systems (pp. 1712-1714).

**Theoretical Claims:**

There is no novel theoretical claim in this paper. All mathematical formulations to the problem are well-defined.

---

> ### Author Rebuttal · Authors · 2025-03-31
>
> We thank the reviewer for your constructive feedback.
>
> **Q1. Concerns about the trajectory mirroring strategy.**
>
> A1.  Our trajectory mirroring strategy is a data preprocessing method specifically designed for the AHT, not a data augmentation method. It works by sequentially designating different agents as the ego agent in a given trajectory. As a result, the combinations of the remaining teammates change, increasing the diversity of teammate sets. This approach not only enhances data efficiency but also provides a wider variety of teammate combinations for training. Through ablation experiments, we have demonstrated that this preprocessing method improves algorithm performance without incurring additional computational overhead. If you believe that other augmentation methods could provide meaningful comparisons, we are open to considering further experiments, but our current focus is to showcase the effectiveness of this method in AHT.
>
> **Q2. Concerns about the teammate's diversity and teammate heterogeneity.**
>
> A2. Thank you for your insightful feedback. To clarify, the robustness of TAGET is focused on its generalization to unseen teammates, especially in terms of adapting to diverse and unknown behaviors. In our experiments (Fig. 3), we evaluate performance using diverse sets of previously unseen teammates to test this generalization. We have added the cross-play experiments between different populations, the cross-play matrix can be seen in https://anonymous.4open.science/r/materials-E410/cross-play-matrix.md.
>
>
> **Q3. Concerns about baseline**
>
> A3. Thanks for this insightful feedback! We agree that including classic offline RL methods like CQL [1] and QT [2] would provide a more comprehensive benchmark suite. We’ve now added CQL as a baseline. **TAGET outperfoms CQL** across all environments, which can be found in https://anonymous.4open.science/api/repo/materials-E410/file/comparison.pdf.
>
> **Q4. Concerns about evaluation metrics.**
>
> A4. There is currently no specific metric for measuring teammate adaptability.  However, achieving a higher return generally indicates better cooperation with teammates in a game. To address this concern, we could consider comparing our agent’s adaptability with the Oracle Upper Bounds suggested by Reviewer cnPV, which would provide a meaningful comparison of our approach in terms of teammate adaptation.
>
> **Q5. Concerns about the novelty.**
>
> A5. Thank you for your comment. While we build on existing research such as Decision Transformer and return-to-go estimation, previous methods are not directly applicable to the novel problem of offline AHT. To address this, we introduce TA-RTG and TA-Goal to handle the unique challenges of offline AHT, where agents must infer and adapt to the behaviors of unseen teammates. This teammate-aware approach enables better generalization in AHT settings, differentiating our method from prior works. We believe this extension of offline RL into the AHT domain is a key innovation.
>
> **Q6. Concerns about the direct ablation for TA-RTG.**
>
> A6. Thank you for your suggestion. We have added direct ablation experiments for TA-RTG, and the results can be found in https://anonymous.4open.science/api/repo/materials-E410/file/ablation_new.pdf. The experimental results show that **TA-RTG outperforms traditional RTG**, as it is able to capture changes in teammates' strategies and adapt to unknown teammates.
>
> **Q7. Confusing figures and lack of detailed descriptions**
>
> A7. Thank you for your valuable feedback. We acknowledge that some figures, such as Figure 3, could benefit from clearer explanations. We will make sure to correct these issues in future revisions.
>
> **Q8. Concerns about the supplementary material.**
>
> A8. Thanks for your useful suggestion.  We commit to including these additional experiment details, such as detailed hyperparameter settings, neural network architecture, the method for computing standard deviation and the computational time of each algorithm, in the Supplementary Material of the revised version. Furthermore, we will also open-source our code.
>
> * [1] Kumar, A., Zhou, A., Tucker, G., & Levine, S. (2020). Conservative q-learning for offline reinforcement learning. Advances in neural information processing systems, 33, 1179-1191.
> * [2] Kalashnikov, D., Irpan, A., Pastor, P., Ibarz, J., Herzog, A., Jang, E., ... & Levine, S. (2018, October). Scalable deep reinforcement learning for vision-based robotic manipulation. In Conference on robot learning (pp. 651-673). PMLR.

---

> > ### Comment · Reviewer_TMEP · 2025-04-08
> >
> > Thank you for your response. Although the additional experiments presented by the authors are somewhat helpful in supporting the conclusions, I remain concerned that the work lacks sufficient theoretical backing to qualify as a strong empirical contribution. Consequently, I have decided to maintain my current evaluation.

---

### Official Review · Reviewer_cnPV · 2025-03-13

**Overall Recommendation:** 3

**Summary:**

This paper addresses ad hoc teamwork in the offline setting. It proposes a method called TAGET, which is based off the decision-transformer architecture and learns from a dataset of offline cooperative multi-agent interactions. It has a couple of main ideas. First, trajectory mirroring: where one agent is sampled to be the ego agent, and the remainder are teammates to be modeled. Second, agent modeling - where the teammate returns to go and a latent variable corresponding to teammate subgoals is modeled. TAGET is evaluated on Predator Prey, Level-Based-Foraging, and Overcooked, and compared against the Decision-Transformer, Prompt-DT, offline ODITS, and offline LIAM. Overall, it improves against baselines on all tasks, in terms of generalization to unseen teammates.

**Claims And Evidence:**

The train and test set of teammates were generated using the same MARL algorithm (Soft Value Diversity). The authors claim, “Each checkpoint represents a distinct joint strategy, capturing a diverse range of behaviors and interaction patterns.” (L282) However, no empirical evidence is presented to back up this claim about a crucial aspect of the experimental design. I would like to see cross-play matrices demonstrating how well different checkpoints can coordinate with each other, focusing especially on whether training teammates can coordinate with testing teammates.

Typically, AHT papers will also generate test-sets using hand-coded heuristic policies, which should be easy to code up / get from other papers for LBF, Overcooked, and Predator-Prey. The analysis would also be stronger if the authors tested against heuristic policies that cover a wide range of teammate behavior.

**Essential References Not Discussed:**

No

**Experimental Designs Or Analyses:**

I have concerns about the strength of the baselines. While TAGET outperforms baselines in all tasks (Fig. 3), many of the proposed baselines are simply not designed for offline ad hoc teamwork, and look like they are entirely failing to learn in most tasks.

- Decision Transformer - offline single-agent RL

- Prompt DT - offline single-agent RL

- ODITS - online AHT

- LIAM - online AHT

I do not think it’s surprising that LIAM and ODITS fails to learn in the offline setting, and that the DT and Prompt DT fail to generalize to the multi-agent setting/unseen teammates (a characteristic of the AHT problem).

While I understand that the authors have proposed the first (to my knowledge) offline AHT method, and therefore there is no other offline AHT baseline, I wonder if they can compare to a method that is closer to the intended setting? For example, why not compare against MADT (Meng et al. 2021), which is an offline multi-agent decision transformer?

I would also be better able to understand how well TAGET is performing if the authors could add some oracle upper bounds to the results plots. For example, perhaps the average self-play score of the test teammates?

**Methods And Evaluation Criteria:**

Evaluation method follows standard protocol in AHT.

**Other Comments Or Suggestions:**

Missing word at Line 33, right column

**Other Strengths And Weaknesses:**

- Strengths

    - This paper fills a hole in the AHT literature

    - The paper has been well-contextualized w.r.t. related work.

    - The ablation study shows that each component of TAGET contributes positively to the final performance

- Weaknesses

    - Hard to understand how strong the method is due to lack of good baselines. The baselines fail because they were not designed for the problem setting, and no oracle upper bounds are presented to contextualize perforamnce.

    - Lack of information about the diversity of the traning and test teammates

    - Overall, the experimental analysis is a little sparse. Some additional empirical questions could be considered, which I have put in the “Questions” section.

**Questions For Authors:**

- How many transitions are included in the offline interaction dataset for each task?
- How does TAGET’s performance change as the size of the offline dataset changes?

**Relation To Broader Scientific Literature:**

The paper is mostly well-situated in the literature, and cites relevant papers that i am aware of. However, I did notice that the authors cited Durugkar et al 2020 as an AHT paper, when it is not. Durugkar et al. 2020 is closer to a decentralized MARL paper.

**Theoretical Claims:**

N/A

---

> ### Author Rebuttal · Authors · 2025-03-31
>
> Thank you for your kind suggestions and helpful feedback!
>
> **Q1: Concerns about teammate diversity.**
>
> A1.  We apologize for the confusion about the teammate generation. First and foremost, Soft Value Diversity (SVD) is not a MARL algorithm but a framework specifically designed to generate diverse policies. SVD maximizes the differences in value estimates across observation-action pairs between different teams [1]. This design ensures diversity between training and testing policy sets. As a result, we did not conduct separate experiments to demonstrate the diversity of teammate sets, since this diversity is inherent to the SVD framework.
> To assuage your concerns, we've added the **cross-play experiments** between different populations, demonstrating significantly lower cooperation efficiency between populations compared to within-population cooperation, empirically validating that the policies represent truly distinct strategies. In cross-population cooperation experiments, each population sequentially designates one agent as the ego agent to collaborate with members from other populations. The matrix notation (Row1, Col3) specifically denotes an experimental configuration where Population 3's designated ego agent interacts with teammates from Population 1, establishing a systematic evaluation framework for inter-population coordination capabilities. As shown in the cross-play matrix(https://anonymous.4open.science/r/materials-E410/cross-play-matrix.md), in each row of the matrix, the diagonal positions behave best, which means that **strategies between populations don't work well in coordinating**, validating the **diversity** of our test teammate sets.
>
> **Q2. Concerns about the baseline and oracle upper bounds.**
>
> A2. Thanks for your raising the issue about baseline selection. As there are no existing methods specifically designed for offline AHT, our paper addresses this gap by proposing the first dedicated solution. To provide a reasonable comparison, we adapted methods from offline RL and online AHT.  We’ve now added MADT [2] as a baseline, as you suggested. Our results show that **TAGET outperforms MADT**(https://anonymous.4open.science/api/repo/materials-E410/file/comparison.pdf). Due to time constraints, we only tested on the teammate test set of size 4, and we will fill in the experiments subsequently. It's important to note that standard MARL methods, including MADT, don’t fully align with the offline AHT setting, as they typically assume teammates of the same type. AHT, however, involves an ego agent adapting to diverse teammates. Our new experiments with MADT confirm this mismatch, highlighting the key strength of TAGET in handling teammate uncertainty.
>
> **Oracle Upper Bounds**: We have included oracle upper bounds represented by the average self-play scores of test teammates, which are shown in the cross-play matrix. **TAGET achieves 86.0%, 77.2%, and 95.1% of the oracle upper bounds** in PP, LBF, and Overcooked environments respectively, which demonstrates strong performance considering the challenging nature of AHT with unknown teammates.
>
> **Q3. Questions about experimental analysis.**
>
> A3.Our offline interaction dataset includes ​2,300,000 transitions in PP, 1,430,000 transitions in LBF, and 5,500,000 transitions in Overcooked. While we have not yet conducted a systematic ablation on dataset size variation, our experiments with ​data augmentation ablations have revealed that our method still maintains reasonable performance under reduced data diversity. We will add related ablation experiments in the future. It's worth mentioning that our trajectory mirroring strategy mitigates the negative effect when the diversity of offline datasets is low (shown in Fig. 5).
>
> * [1] Ding, Hao, et al. "Coordination Scheme Probing for Generalizable Multi-Agent Reinforcement Learning." (2023).
> * [2] Meng, Linghui, et al. "Offline pre-trained multi-agent decision transformer." Machine Intelligence Research 20.2 (2023): 233-248.

---

### Decision · Program_Chairs · 2025-05-01

**Decision:**

Accept (poster)

**Comment:**

This paper proposes an approach for learning an ad hoc teamwork policy offline. It does so by making modifications to the decision transformer approach as well as additional pre-processing of the offline data to enable easier identification of teammate goals and capabilities.
The proposed approach seems to be the first to show a viable offline approach for ad hoc teamwork. But I would still advise authors to ensure that they do not use maximalist language such as `DT is uniquely suited for offline AHT` unless the paper is proving this statement as part of its claims.